



Modeling the mechanisms of coastal vegetation dynamics and ecosystem responses to changing

2                                    water levels

Junyan Ding,[1] Nate McDowell[2,3], Vanessa Bailey,[2] Nate Conroy,[4] Donnie J. Day,[5] Yilin Fang,[6]
Kenneth M. Kemner,[7] Matthew L. Kirwan,[8] Charlie D. Koven,[9] Matthew Kovach,[5] Patrick
Megonigal,[10] Kendalynn A. Morris,[11] Teri O'Meara,[12] Stephanie C. Pennington,[11] Roberta B.
Peixoto,[5] Peter Thornton,[12] Mike Weintraub,[5] Peter Regier,[13] Leticia Sandoval,[5] Fausto
Machado-Silva,[5] Alice Stearns,[10] Nick Ward,[13] Stephanie J. Wilson[10]
1.  Department of Biology, Occidental College, Los Angeles, CA, USA, 90041
2.  Biological Science Division, Pacific Northwest National Laboratory, Richland, WA, USA,
11       99352
3.  School of Biological Sciences, Washington State University, PO Box 644236, Pullman, WA,
13       USA, 99164-4236
4.  Earth and Environmental Sciences Division, Los Alamos National Laboratory, New Mexico,
15       USA, 87545
5.  The University of Toledo, Toledo, OH, USA, 43606
6.  Earth Systems Science Division, Pacific Northwest National Laboratory, Richland, WA,
18       USA, 99352
7.  Molecular Environmental Sciences Group, Argonne National Laboratory, Lemont, IL, USA
20       60439
8.  Virginia Institute of Marine Science, College of William and Mary, Gloucester Point, VA,
22       USA, 23062
9.  Climate and Ecosystem Sciences Division, Lawrence Berkeley National Laboratory,
24       Berkeley, USA
10. Smithsonian Environmental Research Center, Edgewater, MD, USA, 21037
11. Joint Global Change Research Institute, Pacific Northwest National Laboratory, College
27       Park, MD, USA, 20740
12. Environmental Sciences Division, Oak Ridge National Laboratory, Oak Ridge, TN, USA,
29       37830
13. Marine and Coastal Research Laboratory, Pacific Northwest National Laboratory, Sequim,
31       WA, USA, 98382

Corresponding author: Junyan Ding, jding@oxy.edu





**Abstract**
Coastal forests are increasingly experiencing mortality due to inundation by fresh- and
seawater, leading to their replacement by marshes. These shifts alter vegetation composition,
biogeochemical cycling, carbon storage, and hydrology. Using a hydraulically enabled
ecosystem demography model (FATES-Hydro), we conducted numerical experiments to
investigate the mechanisms behind inundation-driven forest loss and the ecosystem-scale
consequences of forest-to-marsh transitions. We compared mortality processes and their effects
across broadleaf and conifer trees at two coastal sites—Lake Erie (freshwater) and Chesapeake
Bay (saline).
Our simulations show that hydraulic failure, driven by root loss under prolonged flooding,
is the primary mortality mechanism across both tree types and sites. Forest replacement by marsh
reduced ecosystem-scale leaf area index (LAI), gross primary production (GPP), transpiration,
and deep soil water uptake in conifer forests, while broadleaf forests experienced smaller
changes due to lower initial LAI and greater marsh compensation. Marsh invasion occurred
following canopy thinning driven by tree mortality. These findings suggest that, under similar
root loss, hydraulic failure dominates coastal tree mortality regardless of species or water type,
with denser forests experiencing stronger ecosystem impacts. Our study identifies key mortality
mechanisms and offers testable hypotheses for future empirical studies on coastal vegetation
change.






## 1 Introduction:


Shoreline vegetation provides important ecosystem functions (Barbier et al., 2011; Mitsch et
al., 2015). Coastal forests can be more productive than adjacent upland systems (Tagestad et al.,
2021), have larger carbon stocks than the marshes replacing them (Smith et al. 2021), mitigate
storm-driven erosion (Arkema et al., 2013; Spalding et al., 2014), and provide habitat for a wide
variety of animals (Duarte et al., 2013; Barbier, 2013; Mitsch et al., 2015). Coastal ecosystems
are experiencing rapid increases in tree mortality due to changing water levels in both fresh- and
seawater systems (McDowell et al. 2022). Globally, sea-level rise and flooding threaten coastal
forests, with potential habitat losses ranging from less than 10% with a 1-meter rise to as much
as 55% under a 3-meter rise scenario (Ury et al., 2021). Increasingly variable freshwater levels in
lakes and accelerating sea level rise (SLR) are anticipated with climate change (Varekamp et al.,
1992; Mimura, 2013; Theuerkauf et al., 2019; Kayastha et al., 2022; Saber et al., 2023). Varying
water levels induce large changes in species composition, including shifts from forest to marsh,
driven in part by tree mortality (Keddy and Reznicek, 1986; Hudon, 1997; Frieswyk and Zedler,
2007; Wilcox, 2004; Wilcox and Nichols, 2008).
Soil hypoxia and salinity are key drivers of tree mortality under increasing inundation.
Prolonged increases in hypoxia and soil salinity reduce root hydraulic conductance and promote
root loss (Colmer and Flowers, 2008; Pezeshki, 2001). Elevated soil salinity also reduces soil
water potential (Boursiac et al., 2005), thereby reducing the soil-to-root water potential gradient
that drives water movement into roots. Together, these belowground impacts of rising hypoxia
and salinity reduce whole-plant hydraulic conductivity (López-Berenguer et al., 2006; Nedjimi,
2014), subsequently increasing the likelihood of xylem embolism and mortality from hydraulic
failure (McDowell et al., 2022). These reductions in whole-plant conductivity can promote
carbon starvation through declining stomatal conductance (Orsini et al., 2012; Sperry et al.,
2016) and leaf loss (Munns and Termaat, 1986; Wang et al., 2019; Zhang et al., 2021b).
Increased salt concentrations inhibit potassium accumulation in guard cells, also promoting
stomatal closure (Clough and Simm, 1989; Perri et al., 2019). These photosynthetic constraints
can be exacerbated by foliar ion toxicity that impairs photosynthetic biochemistry (Ball and
Farquhar, 1984; Delatorre-Herrera et al., 2021; Munns, 2005; Suárez and Medina, 2006; Li et al.,
2021; Yadav et al., 2011).



Physiological impacts from hypoxia and salinity can also vary with the frequency and
duration of inundation and with interspecific variation in physiological traits. Freshwater systems
experience variable inundation across seasons and years (Fig. 1) and may have opportunity to
recover from inundation, whereas SLR induces a chronic rise in inundation that can reduce
recovery (Taherkhani et al., 2020; Thiéblemont et al. 2023). Interspecific trait differences such as
in xylem vulnerability to cavitation can also influence the degree of mortality (Niknam and
McComb, 2000; Lukac et al., 2000; Sairam et al., 2008; Acosta-Motos et al., 2017; Zhao et al.,
2020; McDowell et al. 2022). Mortality mechanism tests have not addressed inundation
dynamics and interspecific variation.
Coastal tree mortality has large impacts on ecosystem function (Kirwan and Gedan, 2011).
Tree mortality reduces forest biomass and leaf area (Chen and Kirwan, 2022), which in turn
provides a light environment favorable for the expansion of marsh vegetation (Shaw et al., 2022;
Sward et al., 2023). This results in transpiration and carbon uptake and storage shifting from
trees to marsh plants as forests retreat, with net reductions in fluxes and storage depending on the
degree of compensation by the invading marsh (Smith et al., 2021; Davidson et al., 2018; Zhou
et al., 2023). However, the effects of tree death and marsh invasion on ecosystem-scale fluxes
and their underlying mechanisms are poorly known (Kirwan et al., 2024).
In a recent study, Ding et al. (2023b) incorporated the effects of salinity and hypoxia on root
loss, hydraulic function, and photosynthesis into the process-based ecosystem demographic
model FATES-Hydro to simulate physiological responses of conifer trees to seawater exposure.
They applied the model to three coastal conifer forests in the U.S., finding that both hydraulic
failure and carbon starvation contribute to tree mortality, with the dominant mechanism
depending on the rate and duration of salinization. Rapid exposure favored carbon starvation,
while chronic inundation led primarily to hydraulic failure.
Building upon Ding et al. (2023b), which focuses on physiological responses of conifer trees
to seawater exposure, here we extend the investigation to broadleaf deciduous trees, allowing
evaluation of how functional traits such as photosynthetic capacity, morphology, and phenology
influence mortality outcomes under similar belowground stress conditions. Because the two tree
functional groups have very different physiology, morphology and phenology, we specifically
examine to which extent the difference in these traits can affect the response of trees to hypoxia





and salinity if root loss are the same. We also incorporate a freshwater coastal system (Lake
Erie) alongside a saline coastal site to compare how hydrologic context shapes vegetation
transitions. We further assess ecosystem-scale consequences of tree mortality, including its
impact on vegetation composition, and ecosystem fluxes. We conducted numerical experiments
using Ding et al.(2023b) FATES-Hydro with a reciprocal design where the two tree functional
groups were simulated for both the freshwater and saline coasts to partition the role of species
from their environment. We expect the broadleaf deciduous trees will be more susceptible to
carbon starvation particular at saline coastal than conifers.
2. **Methods**
2.1 Study areas

The first site is located on the south shore of Lake Erie (the LE site) in Ohio, USA (41.48°N,

83.06°W) (Fig. S1). The area has warm and humid summers and cold winters. The LE site is
dominated by the broadleaf species shellbark hickory (*Carya laciniosa*) and swamp white oak
(*Quercus bicolor*). Lake Erie is part of the Great Lakes of North America, a series of
interconnected large freshwater lakes. The water level of Lake Erie is subject to daily, seasonal,
interannual, and decadal variation resulting from the complex interactions between climate,
bathymetry, and the water levels of the upper lakes (Burlakova et al., 2014). Increases in the
water levels in Lake Erie have resulted in extensive coastal tree mortality (Sippo et al., 2018;
Theuerkauf and Braun, 2021).

The second site is located on the Chesapeake Bay (CB) in Maryland, USA (38.5°N, 76.3°W;

Smith and Kirwan 2021) (Fig. S1). The CB site is dominated by coniferous loblolly pine (*Pinus
taeda*) forests, and the climate is characterized by warm, humid summers and cool winters. The
CB region is a hotspot for sea-level driven coastal forest retreat, driven in part by the extensive
low lying coastal plain topography (Schieder et al., 2018; Chen and Kirwan, 2022). In the 20[th]
century, relative SLR rates in the CB region (~3 to 6 mm per year) are approximately two to
three times faster than the global average (Sallenger et al., 2012; Ezer and Corlett, 2012).

In 2022, two forested study plots were established within the dying shoreline forests and in

the neighboring unflooded uplands at each site. Stand level measurements include tree density,
diameter at breast height (DBH) and water table depth. Tree-specific measurements were taken
from eight live trees in each plot, which included growth, non-structural carbohydrates,





continues hourly sap flow, leaf gas exchange, leaf photosynthetic capacity, and hourly leaf water
potential (LWP) including predawn and mid-day LWP of a given day during the growing season.
These eight live trees were bored to obtain tree cores for measurements of ring width growth
along with eight dead trees located at the shoreline. We follow exactly the same measurements
and sample processing as described in Zhang et al. (2021), Wang et al. (2020). Here, we
benchmarked the FATES-Hydro model against empirical data collected at each site.

2.2 Numerical experiments with the FATES-Hydro model
We conducted numerical experiments at both the LE and CB sites using a newly developed
version of the ecosystem demography model, FATES-Hydro (Ding et al. 2023b), which
represents the physiological impacts of hypoxia and salinity on plants and consequent changes in
root conductance and mortality. Here we describe this version of FATES-Hydro, its
parameterization and benchmarking for this study, and the design of the numerical experiments.
*2.2.1 Description of FATES-Hydro*
The Functionally Assembled Terrestrial Ecosystem Simulator (FATES) is a physiology-based
vegetation demographic model that simulates cohort-scale dynamics for different plant
functional types (PFTs) (Fisher et al., 2018; Koven et al., 2020). FATES-Hydro is a version that
integrates the plant hydraulics and their coupling with photosynthesis (Ding et al., 2023a). In
FATES-Hydro plant transpiration is the product of whole-plant leaf area and the transpiration
rate per unit leaf area (J), which itself is the product of stomatal conductance and vapor pressure
difference from leaf intercellular spaces to bulk atmosphere. The hydro-dynamic module
represents a plant's roots, xylem, and foliage as a variably porous media (Sperry, Adler,
Campbell, & Comstock, 1998) with conductance and capacitance changing in response to tissue
water potentials dictated by the pressure-volume (P-V) curve and the pressure-conductance
(vulnerability) curve (Manzoni et al. 2013, Christoffersen et al. 2016). Stomatal conductance is
modified from the Ball-Berry model (Ball et al. 1984, Oleson et al. 2013, Fisher et al. 2015) with
a further constraint of leaf water potential through a water stress index $\beta t$, defined by a function
of the ratio of the leaf water potential to the leaf water potential of half stomatal closure (P50gs)
(Christoffersen et al. 2016).  The soil column is divided into a given number of layers. The
proportion of roots in each layer is calculated from Zeng's (2001) two parameter power law



function. Water flow from each soil layer within the root zone into the plant root system is
calculated as a function of the hydraulic conductance determined by root biomass and root traits
such as specific root length, and the difference in water potential between the absorbing roots
and the rhizosphere. Additional technical details and parameter sensitivity analysis can be found
in the technical notes (FATES Development Team, 2021, https://fates-users-
guide.readthedocs.io/projects/tech-doc/en/latest/index.html) and publications (e.g. Koven et al.,
2020; Xu et al., 2023, Ding et al., 2023a, 2023b, Robbins et al, 2024 ).
The version of FATES-Hydro used in this study also includes representation of the
mechanisms by which soil hypoxia and salinity impact tree physiology and mortality (Ding
2023b). The complete description of these new developments can be found in Ding et al.
(2023b). Below we describe the root loss function because this is the key component in this
study.
The root loss function (kr$_{red}$), expressed as the proportion of the root conductance under
normal condition, is composed of hypoxia reduction ($kr_{red,sat}$) and the salinity reduction ($kr_{red,sal}$),
expressed as:
$$kr_{red} = kr_{red,sat} \cdot kr_{red,sal}$$  Eq.1
Each term varies between near zero to 1, with near zero means no roots and 1 means the
roots grow as normal.
The saturation reduction ratio is given as:
$$kr_{red,sat} = \frac{1}{1+b \cdot e^{ks(x-x_0)}}$$  Eq.2
where *b* and *ks* are the scaling parameters that determine the rate of fine root loss from
saturation; $x$ (hours) is the total duration of the volumetric soil water content [m3/m3] exceeds
90% saturation over a defined previous period of $x_0$. Biologically, parameter *b* and *ks* can be
used to represent how well the root system of the trees are adapted to waterlogging condition
(Fig S2).
The salinity reduction ratio is given by a salinity cumulation term (acc_sal) as:
$$kr_{red,sal} = exp(-kc * acc_{sal})$$  Eq.3a





$$acc_{sal} = max[\,0, \sum (Sal_{soil,t} - Sal_{cr})]$$ Eq.3b
where $kc$ is a parameter determining the rate of fine root loss due to salinity; $acc_{sal}$ (PSU) is the
cumulative salinity since simulation starts, $Sal_{soil,t}$ is the soil salinity at a given time, $Sal_{cr}$ is the
critical soil salinity beyond which salinity starts to negatively affect root mass.

*2.2.2 Parameterization and benchmarking*
For the pine and the salinity induced root loss rate, the parameters used in Ding et al. (2023)
were used in this study. For the broadleaf trees at Lake Erie, the parameters used in FATES-
Hydro were either from field observations or obtained from the TRY trait database (Kattge et al.,
2011) when field observation were not available (Table S1). $V_{cmax}$ was estimated from A/Ci
curves and then adjusted within the observed range so that the simulated hourly $A_{net}$ matched the
fitted line of observed values (Fig. S2). $P50_{gs}$ was adjusted within the ranges of the temperate
broadleaf trees from the TRY database so that the simulated hourly leaf water potential matches
the fitting curved based on measured hourly LWP (Fig. S2).
The allometry parameters that define the relationships between diameter at breast height and
total tree height, sapwood area, total woody biomass, and total leaf biomass were estimated
based on values of the Biomass and Allometry Database (BAAD) (Falster et al., 2015). The
complete list of parameters can be found in the GitHub repository:
https://github.com/JunyanDing/FATES_COMPASS.
To calibrate the plant hydraulic parameters, $K_{max}$ and xylem vulnerability curves, we first
adjusted the parameters within the observed range of values for our broadleaf species (Kattge et
al. 2011) from TRY database of the site to be close to measured hourly values and then we
adjusted the phenology parameters so that the decline of simulated sap flow matched the
observed pattern (FigS4 top panel). For saturation induced root loss function, based on previous
studies (Islam and Macdonald 2004, Aroca et al. 2012, Karlova et al. 2021) and unpublished
experimental observations (B. Wolf unpublished data), we set $x_0$ to 120 hours (5 days) for this
study. We then adjusted the root loss parameter $ks$ so simulated sap flow was close to the mean
of the observed values (FigS4 bottom panel) at the shoreline location with observed water table



depth. The simulated average daily sap flow at upland and shoreline locations matches well with
observations (Fig. 2). While perfect agreement is not expected, we find that the model
reasonably captures the seasonal pattern and interannual variability of sap flow.
For the marsh grass, we use the biochemical and physiological parameters from O'Meara et
al. (2021) and the allometry (height and total leaf area to stem width ratio) was estimated based
on GCReW data (https://serc.si.edu/gcrew/data). Because marsh plants are annual or bi-annual,
phenology and maximum density (number of individuals per ground area) that control the
variation of total leaf play more important role in marsh ecosystems than plant physiology. We
specifically calibrated these parameters. The phenology parameters are estimated based on the
NDVI values from Yaping et al. (2022) and the 2022 field measurement of sap flow at the LE
upland site. The phenology parameters are same for both marsh plants and broadleaf trees,
meaning marsh plant and broadleaf trees have same leaf on and off times. The NDVI values at
the neighboring wetlands indicate the marsh grass system had an LAI ~2 ($m^2$ $m^{-2}$) during the
peak growing season. We further constrained the maximum density of the marsh to a growing
season LAI of 2 $m^2$ $m^{-2}$. To calibrate the broadleaf trees, we ran the model at LE site in 2022
initialized by the inventory data, then compared the model output with the field observations of
photosynthesis, leaf water potential (Fig S2), and sap flow (Fig S3). We confirmed that the
simulated growth rates over the 30-year simulation period fell within the observed range of tree
ring widths measured from tree cores (Fig S4). Note, our goal was to assess if simulated growth
rates were within the range of observations to enable hypothesis tests, not to reproduce observed
interannual growth variability at the individual tree level.
*2.2.3 Setup of numerical experiments*
The numerical experiments involved three plant functional types (PFT): broadleaf deciduous
tree, evergreen conifer tree, and herbaceous marsh plants. We will call them broadleaf tree,
conifer tree, and marsh plants hereafter. Each simulation was constructed either as broadleaf
tree-marsh or as conifer tree-marsh combination. We first simulated the LE site using its native
vegetation, the broadleaf trees and marsh, and the CB site using its native vegetation, conifer
trees, and marsh. We then swapped the tree types between sites, such that the broadleaf forest
was simulated at CB and the conifer forest at LE. The simulations of virtual forests at each site
allowed investigation of how different forest types may respond to the differing drivers, i.e., with



and without salinity. For both experiments we also examined the ecosystem-scale consequences
of forest loss, namely on total evapotranspiration and photosynthetic carbon fluxes.

The simulations were driven by the University of East Anglia Climatic Research Unit

(CRU) Japanese Reanalysis (JRA) meteorological product (CRUJRA) (University of East Anglia
Climatic Research Unit; Harris, 2019) for 1990 through 2019. The simulations were initialized
with inventory data of upland locations for the trees at both sites (Fig. S1). We used the observed
inventory data of *Carya spp.* at LE for both CB and LE initialization and observed inventory data
of *Pinus spp.* at CB for both CB and LE initialization The simulated marsh colonization was
from external seed supply at the first year, then from both external seed supply and local
reproduction afterward.

Daily soil salinity and water table depth at CB and water table depth at LE were used as

external driving factors. We used empirically estimated soil salinity at CB, by regression based
on open water level and salinity (Ding et al. 2023b). To estimate water table depth at LE from
1990 to 2019, we obtained the water level of LE at the station in Cleveland
(https://tidesandcurrents.noaa.gov/inventory.html?id=9063063). We fit a linear correlation
between the station water level and the observed water table depth at the LE shoreline location in
2022 (Fig 1a), then used this linear model to estimate the daily water depth from 1990 to 2019
via the station water data (Fig 1a). During the simulation period, two floods occurred at LE
(1997–1999 and 2016–2019); at CB, salinity slightly rose around 2002 and 2009, followed by a
constant increase after 2012 (Fig 1d and 1e).

Root loss was driven by soil hypoxia (indexed by the duration of saturated water content) for

LE, and both soil hypoxia and salinity for CB. At LE, the root loss was calculated based on water
table depth. The parameters that govern root loss were calibrated based on measured sap flow,
leaf water potential, and loss of plant hydraulic conductance (S1). At CB, the additive root loss
was estimated from soil salinity because soil salinity is highly coupled with hypoxia at CB, and
we used the parameter values from Ding et al. (2023b). Root loss was simulated similarly for
both species to explicitly examine the extent to which differences in phenology, leaf and stem
physiology, and the lack of species-specific information on hypoxia and salinity tolerance may
result in different mortality patterns.



3. **Results of numerical experiment**
3.1 Inundation impacts on tree physiology and mortality
Inundation had similar impacts on the broadleaf and conifer trees at both LE and CB. Root
loss increased over time as water levels rose (Fig 2a, b). However, root loss differed between
sites, particularly after 2015, due to the differences in inundation dynamics. At LE, root loss
increased each year as water levels rose, but root growth during periods of low water levels
allowed partial recovery. In contrast, the chronic increase in inundation and soil salinity at CB
after 2015 led to ongoing root loss because there was less seasonal variation in water level (Fig.
1c and d), and hence no opportunity for recovering root biomass. This difference in root loss
between sites resulted in sustained reductions in hydraulic conductance ($k/k_{max}$) at CB after 2015,
whereas both species at LE exhibited some recovery each year (Fig. 3c, d).
Despite these site level differences, both LE and CB experienced severe declines in $k/k_{max}$
by the end of the simulation period. Non-structural carbohydrates (NSC) showed only slight
variation over the simulation period, suggesting that loss of hydraulic conductance was the
primary process underlying mortality (Fig. 3e, f). This is consist with observed %NSC at Lake
Erie site (table S2). Mortality of both the broadleaf and conifer trees increased after 2015, with
higher mortality at CB than LE (Fig. 3g, h). Increasing root loss was associated with declining
$k/k_{max}$ (Fig. 4a, b) and increased whole-tree mortality (Fig. 4c, d), with no difference between
species. The declines in $k/k_{max}$ were strongly associated with increased mortality (Fig. 4).
3.2 Ecosystem consequences of forest loss
*3.2.1 Broadleaf simulations at Lake Erie and Chesapeake Bay*
We examined the ecosystem-scale consequences of mortality on leaf area index (LAI), gross
primary production (GPP), transpiration ($E_t$), and root water uptake with the broadleaf
simulations at both the LE and CB shoreline sites. In these simulations, the loss of leaf area
through inundation-driven tree mortality was compensated by marsh invasion, resulting in stable
ecosystem-scale LAI over time (Fig. 5a, 5b). GPP and $E_t$ showed somewhat similar patterns as
LAI for both sites, again due to marsh invasion as trees died (Fig. 5c-f). GPP was slightly higher
in the shoreline than the upland sites due to the higher GPP of marsh plants, and $E_t$ showed slight
declines below the upland sites (Fig. 5b and 5c). While LAI, GPP, and $E_t$ all showed relative
stability over time, the loss of trees and the marsh invasion led to large changes in the depths of



water uptake at both LE and CB. The change in vegetation dominance associated with tree
mortality led to an increase in shallow water uptake and a decline in deep water uptake (Fig. 5g-
j).
*3.2.2 Conifer simulations at Lake Erie and Chesapeake Bay*
The conifer simulations at the shoreline sites at both LE and CB exhibited different patterns
than the broadleaf species. LAI, GPP, and $E_t$ all declined with tree loss, which was not
compensated for due to limited marsh invasion (Fig. 6a-f). There was no change in shallow water
uptake with changes in vegetation dominance, but there was a large decline in deep water uptake
(Fig. 6g-j). As tree LAI declined with mortality, the increase in shallow water uptake observed
for broadleaf species was not observed for the conifer species, whereas both species exhibited
declines in deep water uptake (Fig. 7).
4. **Discussion**
4.1 Summary of Major Findings
We conducted numerical experiments at two coastal sites to investigate the mechanisms
driving inundation-driven tree mortality and the subsequent ecosystem impacts and how they
differ across tree species and with different inundation regimes. The simulations indicated that
root loss was the dominant step driving mortality via hydraulic failure for both species and at
both sites (Fig. 3 and 4). Replacement of broadleaf trees by marsh resulted in increased LAI and
GPP but reduced $E_T$ at both sites (Fig 5a to 5f). Replacement of conifer trees by marsh result in
reduced LAI, GPP, and $E_T$ at both sites (Fig. 6a to 6f). Transition from forests to marsh shifted
root water uptake from deep to shallow soil layers (Fig 4g, 4h, 6g, 6h). Our numerical
experiments suggest that the same mechanisms caused forest loss at both sites regardless of tree
type, whereas the ecosystem effects from the replacement of forest by marsh differed between
broadleaf and conifer forests. Future empirical studies should be conducted to verify these
findings.
4.2 Tree Level Effects
Root loss can promote tree mortality through hydraulic failure and carbon starvation
(McDowell et al., 2022). Our simulations indicated that hydraulic failure is the dominant process
underlying tree mortality for broadleaf and conifer trees at both sites (Fig. 3 and 4). Root loss





resulted in decreased whole tree hydraulic conductance and subsequently xylem conductivity due
to increased xylem embolism (Fig. 4a and 4b). Overall, this led to an increase in tree mortality
(Fig. 4c and 4d). These effects were consistent between LE and CB, although they exhibited
different temporal patterns due to variation in the inundation regimes. Moreover, similar effects
of root loss on plant hydraulics have been documented in both modeling studies (Li et al., 2022;
Ding et al., 2023b) and field studies (Zaerr, 1983; Pezeshki et al., 1996; Andersen et al., 1984;
Islam and Macdonald, 2004; Aroca et al., 2012; Karlova et al., 2021).

Reduced hydraulic conductivity can lower leaf water potential and causes stomatal closure.

This mechanism can result in reduced photosynthesis and negative carbon balance, resulting in
reduced capability to maintain tissues and defend against insects and pathogens (McDowell et
la., 2022). However, simulated non-structural carbohydrates (NSC) values (Fig. 3e and 3f)
suggest that carbon starvation was not a major process of tree mortality at the shoreline locations
of both study sites and for both tree types. These results were unexpected. We had anticipated
that broadleaf trees might experience greater carbon limitation due to higher leaf area and
photosynthetic demand. However, the simulations demonstrated that hydraulic failure associated
with root loss occurred before significant depletion of NSC, leading to similar mortality
trajectories between the two species. This may be due to the rate of inundation-driven root loss
relative to the rate of NSC decline (Ding et al., 2023b). The LE and CB inundation regimes had
periods when inundation declined and the salinity at CB was relatively low, allowing a low level
of ongoing photosynthesis to replenish their NSC pools. Carbon starvation is a slow process due
to the time required to draw down NSCs, whereas hydraulic failure can occur rapidly (McDowell
et al., 2022).

Simulated $k/k_{max}$ and mortality of broadleaf and conifer trees changed similarly with root

loss (Fig. 4), despite large differences between the two species in leaf economic traits, wood
anatomy, crown allometry, and phenology. This similarity arose because whole-tree $k/k_{max}$ can
only be as high as the lowest $k/k_{max}$ of any pathway between the soil and foliage. Root loss
caused the soil-to-root $k/k_{max}$ to decline dramatically, forcing whole-tree $k/k_{max}$ to equal soil-to-
root $k/k_{max}$. Thus, traits and processes downstream from the roots became less important due to
the dominant role of root loss in promoting hydraulic failure when it becomes severe (Fig. 3c, d,
g, h). Both species are poorly adapted to high levels of hypoxia or salinity, thus root loss was the



critical failure point in tree survival under inundation. Species with root systems adapted to
inundation, such as mangroves, may experience different consequences of increased flooding
that could lead to a larger role of carbon starvation such as through ion toxicity to photosynthesis
and leaf loss (Munns and Termaat, 1986).

This convergence in response does not imply that all species react identically to inundation.

Rather, it reflects that under the modeled conditions, root system failure overwhelms the
contributions of other physiological differences. While these results offer mechanistic insights,
the lack of empirical data on species-specific root adaptations remains a limitation. As such, we
caution against overgeneralization and recommend interpreting these results as hypothesis-
generating rather than conclusive. Future research on the cross-species variation of root loss and
downstream mortality mechanisms will be useful to advance transferable predictive capacity of
coastal vegetation change under increasing inundation.

4.3 Ecosystem-Level Effects Between the Forest Types and Sites

The ecosystem-scale consequences of coastal forest loss result both from the loss of trees

and the invasion of marsh plants. We found differences in the ecosystem consequences
associated with species but not sites. As tree mortality associated with soil inundation and/or
salinity progressed, marsh plants invaded broadleaf systems more rapidly than coniferous
systems, thus resulting in different impacts on GPP and $E_t$. Marsh plants invaded when LAI
declined below 1 $m^2/m^2$ in both systems (Fig. 5a, 5b and 6a, 6b). The conifer system had higher
stand density than the broadleaf system resulting in initial LAI values of ~4 $m^2/m^2$ and ~2 $m^2/m^2$,
respectively, thus a much larger amount of mortality was required in the conifer system for LAI
to decline by 1 $m^2/m^2$; in other words, when canopy opened hereby facilitated marsh invasion
and establishment. The mortality rates were similar for both species (Fig. 3); thus, the differences
in marsh invasion were due primarily to initial stand structure rather than to species composition
or mortality rates *per se.* The declining LAI in the CB with increasing mortality is consistent
with remotely sensed estimates of the normalized difference vegetation index (Chen and Kirwan
2022), and the rates of marsh invasion are consistent with other observations (Kirwan and
Gedan, 2019; McKown and Burdick, 2024). This resulted in increasing GPP in the broadleaf
system because marsh plants have higher photosynthetic capacity than trees (Pan et al., 2020)





and because ecosystem LAI was higher after marsh invasion into the broadleaf system (Fig. 5c,
5d and 6c, 6d). In contrast, the slower invasion of marsh plants into the conifer system caused
ecosystem level GPP to decline with tree mortality.

Replacing broadleaf forest by marsh increased GPP slightly, while $E_t$ declined slightly (Fig.

7c and 7d). This is likely due to the higher water-use efficiency of marsh plants than trees. $E_t$
declined with tree mortality in both systems, but more so in the conifer system (Figs 5e, 5f and
6e, 6f). These changes were associated with increased water uptake from shallow soil layer and
reduced uptake from the deep soil layer due to the shallower roots of the marsh plants (Fig. 8)
(Maitre et al., 1999; Schenk and Jackson, 2002). In contrast, conifer mortality resulted in less
marsh invasion and hence reductions in GPP and $E_t$ (Fig. 7), and water uptake from both shallow
and deep soil layers was greatly reduced due to the loss in conifer roots that were not
compensated by marsh plants (Fig. 8).

4.5 Future Research

Our numeric experiments revealed critical mechanisms regulating vegetation dynamics and

ecosystem impacts in response to SLR. Our study also revealed key next steps to improve our
understanding and model representation. We found that root loss drives large declines in $k/k_{max}$
that result in increasing mortality and marsh invasion. However, we did not directly measure root
distribution, but instead calibrated parameters using observed data on leaf water potential, sap
flow, and the $k/k_{max}$. We then applied these calibrated parameters with the assumption that both
the broadleaf and conifer had similar root responses to hypoxia and salinity. This decision was
based on three factors. First, we had no species-specific data on root conductance and mortality
in response to hypoxia and salinity. Second, both species live at the shoreline margins of their
respective regions, and thus, we assume they are similar in their root responses. Third, the
rooting depths at both sites are shallow, thus rooting depth may not vary significantly across
species. By setting the root loss parameters the same for both tree species, we investigated
whether differences in phenology, leaf and stem physiology traits result in different mortality
patterns between the two species and found that these trait differences have negligible impacts.
Given the critical role of root conductance and survival, field studies to explore root structure
and function are important next steps for coastal systems. Ecosystem manipulation experiments





can be particularly powerful to untangle the impact of hypoxia and salinity on root loss and
subsequent impacts on physiology and survival, enabling cause-and-effect tests that enable
improved predictive understanding of tree mortality in coastal systems (e.g., Hopple et al., 2023).

The role of changing stand structure during ghost forest formation may be an important

factor impacting marsh invasion, the physiology of surviving trees, and the recruitment of new
trees. Changes in light availability as mortality increases aided marsh invasion and promoted
higher photosynthesis and recruitment (Fig 5a and 5b, Fig 6a and 6b) (Kirwan and Gedan, 2019).
The reduction of water uptake from deep layers may promote hypoxia, salinity, and changes in
redox potential and nutrient cycling. With forest cover, high deep-water uptake can enhance
infiltration of rainfall into deep soil layers, which can bring oxygen rich surface water and
nutrients to these deep layers. Reduced infiltration could result in lower dissolved oxygen in
deep soil (Foulquier et al., 2010), increased salinity (Kirwan et al. in review), changed redox
potential (Rubol et al., 2012), and decreased nutrient availability (Burgin et al., 2010; Burgin et
al., 2012) all of which should feedback to limit vegetation growth. Therefore, quantifying the
rate of marsh invasion, survival of remaining trees, and recruitment of tree seedlings is necessary
to identify mechanisms associated with changing light availability and belowground processes.

4.6 Summary

Our numerical experiments indicated that root loss due to coastal inundation served as the

driving force behind the transition from forest to marsh through hydraulic failure-induced tree
mortality. This mechanism resulted in similar physiological consequences for both broadleaf and
conifer trees. However, the transition from forests to marshes led to different ecosystem impacts
between broadleaf and conifer forests due to their different initial LAI. Future research aimed at
enhancing our understanding and representation of the interplay among physiological,
demographic, and stand structural processes in different forest types is necessary for better
predicting vegetation dynamics and ecosystem consequences under SLR.
**Acknowledgments**

JD, NM, BBL, KM, NW, JPM, PR, SCP, MW, TM, PT, and VB were supported by the

Department of Energy, Biological and Environmental Research program project Coastal



Observations, Mechanisms, and Predictions Across Scales (COMPASS). Any use of trade,
product or firm names is for descriptive purposes only and does not imply endorsement by the
U.S. Government. MK acknowledges the support of the U.S. National Science Foundation
(#1654374, #1832221, #2012670).
**Author contributions**
JD and NM designed the study and drafted the manuscript. JD developed the model and
performed simulations. CDK helped with model development. NM, NW, LS, DD, KM, MK, PR,
PZ, HZ, SP, SW, WW, WI, AS, TM, and PT collected field data and performed data analysis.
All the authors contributed to the manuscript.
**Competing interests**
None declared
**Code availability statements**

The data and FATES-Hydro code that support the findings of this study are openly

available in GitHub repository: https://github.com/JunyanDing/FATES_COMPASS or zendo:
https://doi.org/10.5281/zenodo.15116449



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





**Figure caption**
Fig.1 Study area: a) relation between lake depth and soil water table depth in 2022, at the Lake
Erie site transition zone corresponding to the shoreline location in simulations; b) depth of Lake
Erie 1990–2020; c) Estimated water table depth at Lake Erie (LE) shoreline location; d) sea level
and open water salinity of the nearby station of Chesapeake Bay (CPB); and e) Estimated soil
salinity at CPB site
Fig.2 Benchmark root loss function by comparing measured and simulated average daily sap
flow of Aug and Sep. 2022 at upland location and shoreline location at LE site
Fig.3 Simulated tree level variables of shoreline forest at Lake Erie and Chesapeake Bay: a) and
b) monthly mean % live root; c) and d) monthly mean $k/k_{max}$; e) and f) annual %NSC; g) and h)
annual mortality rate
Fig.4 Simulated relation between a) and b) mean growing season $k/k_{max}$ and % live roots of tree;
c) and d) annual mean mortality and % live roots of tree
Fig.5 Ecosystem effects of broadleaf forest at Lake Erie and Chesapeake Bay: a) and b) mean
growing season leaf area index (LAI); c) and d) gross primary productivity (GPP); e) and f)
transpiration ($E_t$); g) and h) root water uptake rate from shallow soil; i) and j) root water uptake
rate from deep soil at shoreline (SH) and upland (UP) locations
Fig.6 Ecosystem effects of conifer forest at Lake Erie and Chesapeake Bay: a) and b) mean
growing season leaf area index (LAI); c) and d) gross primary productivity (GPP); e) and f)
transpiration ($E_t$); g) and h) root water uptake rate from shallow soil; i) and j) root water uptake
rate from deep soil at shoreline (SH) and upland (UP) locations
Fig.7 Change in gross primary productivity (GPP) (a and b) and transpiration ($E_t$) (c and d) with
tree abundance of shoreline forests as indicated by tree LAI at Lake Erie and Chesapeake Bay
Fig.8 Change in water uptake from shallow soil layer (a and b) and from deep soil layer (c and d)
with tree abundance of shoreline forests as indicated by tree LAI at Lake Erie and Chesapeake
Bay



**Figures**

Fig.1

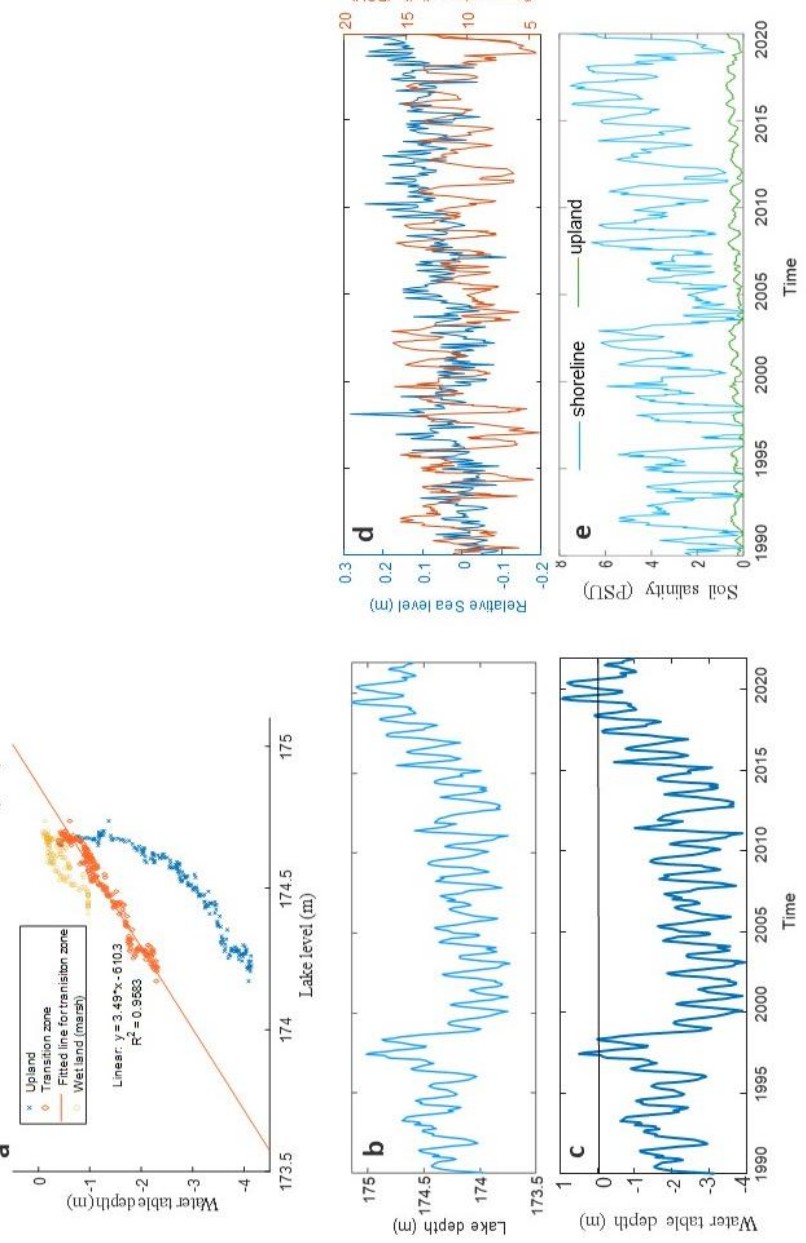



Fig. 2

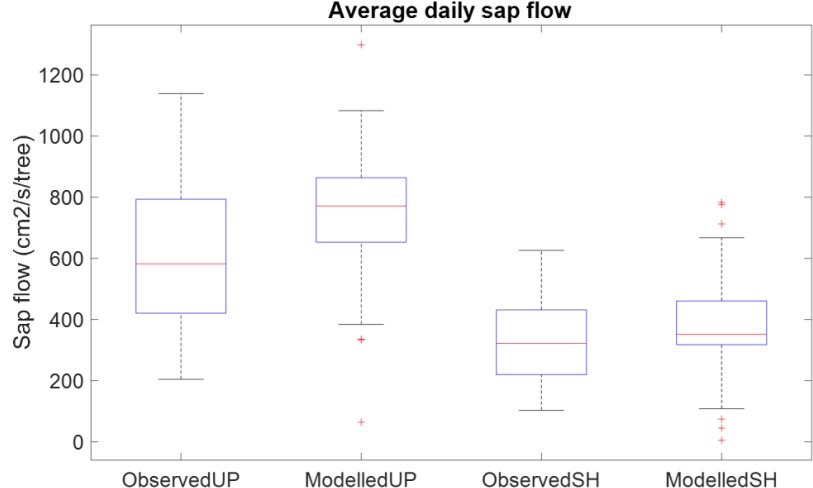



Fig.3

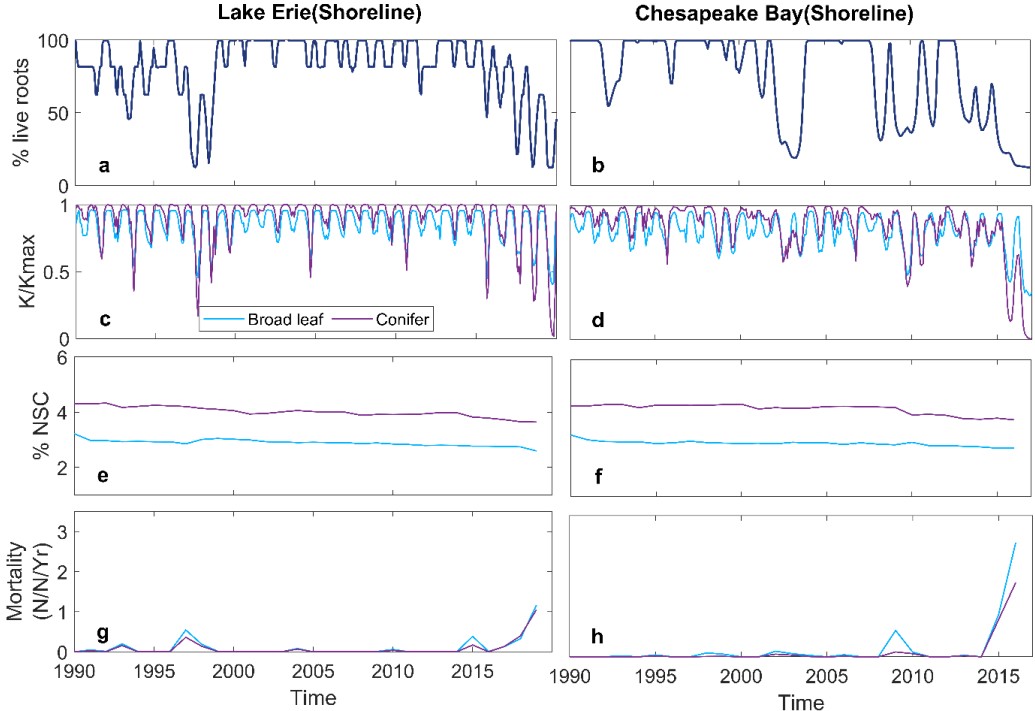





Fig.4

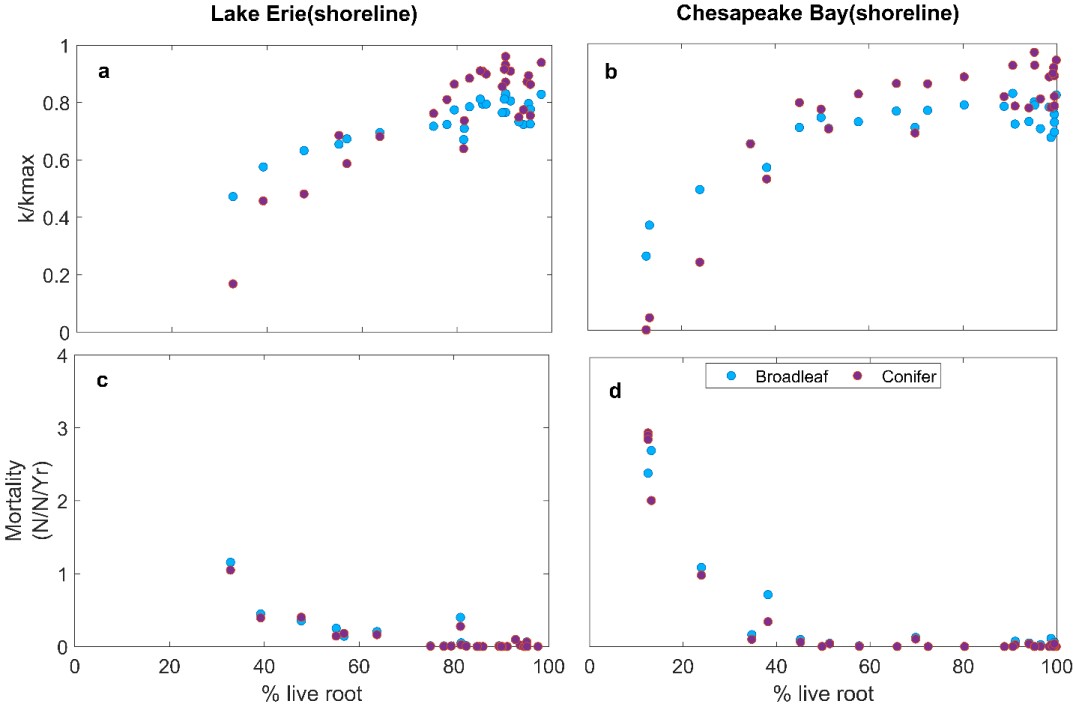



Fig.5

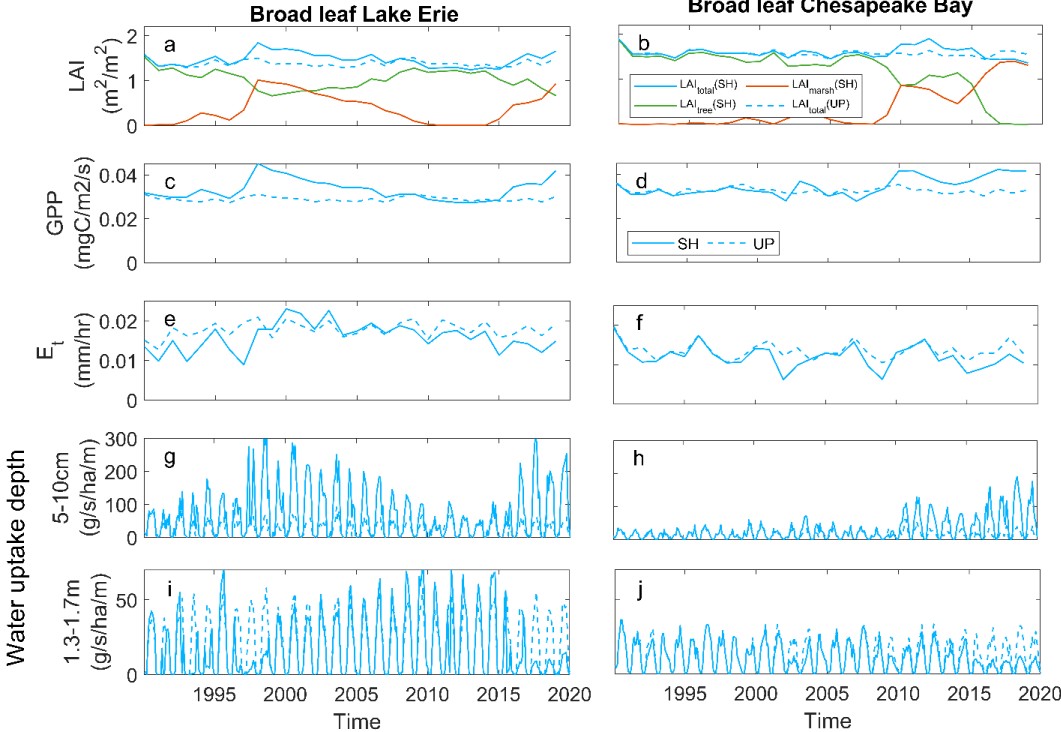




Fig.6

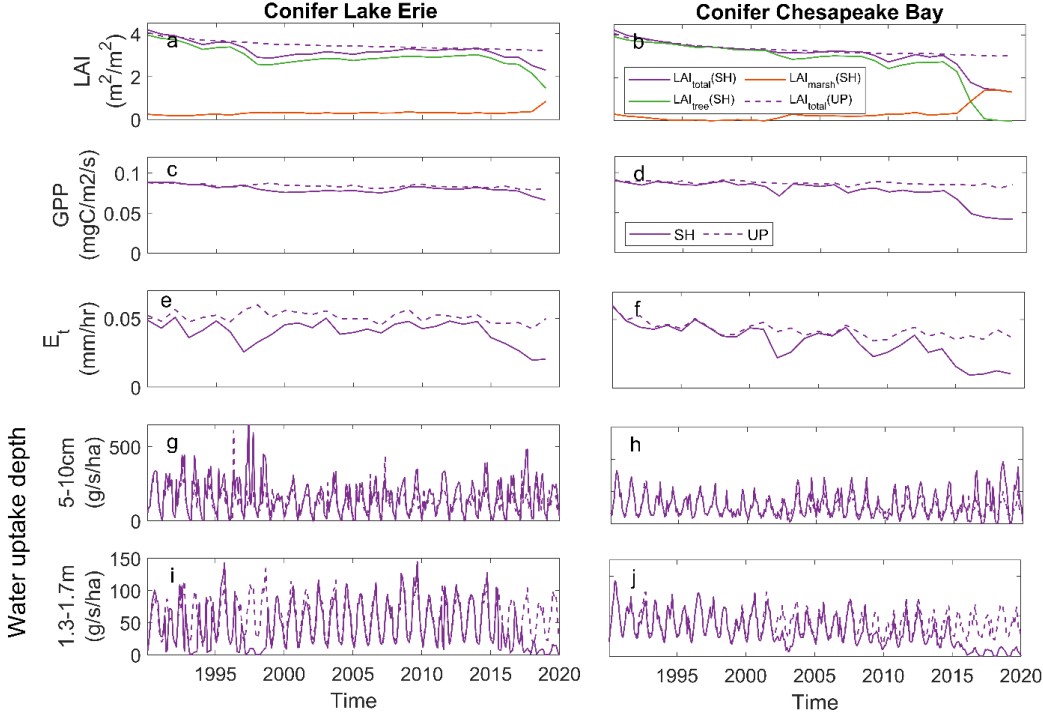





Fig. 7

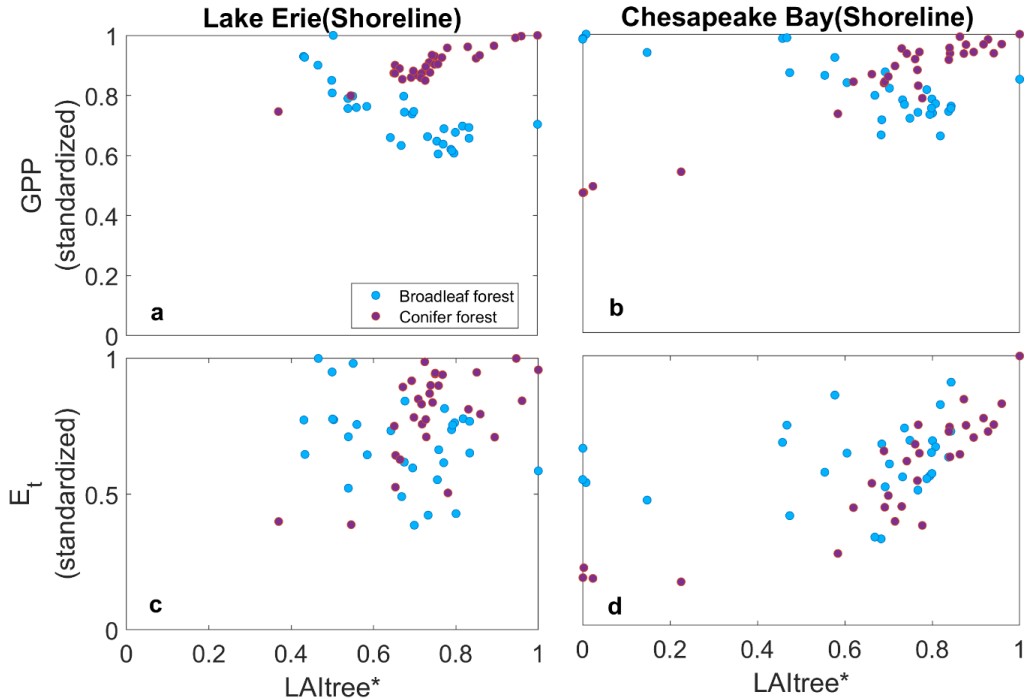





Fig.8

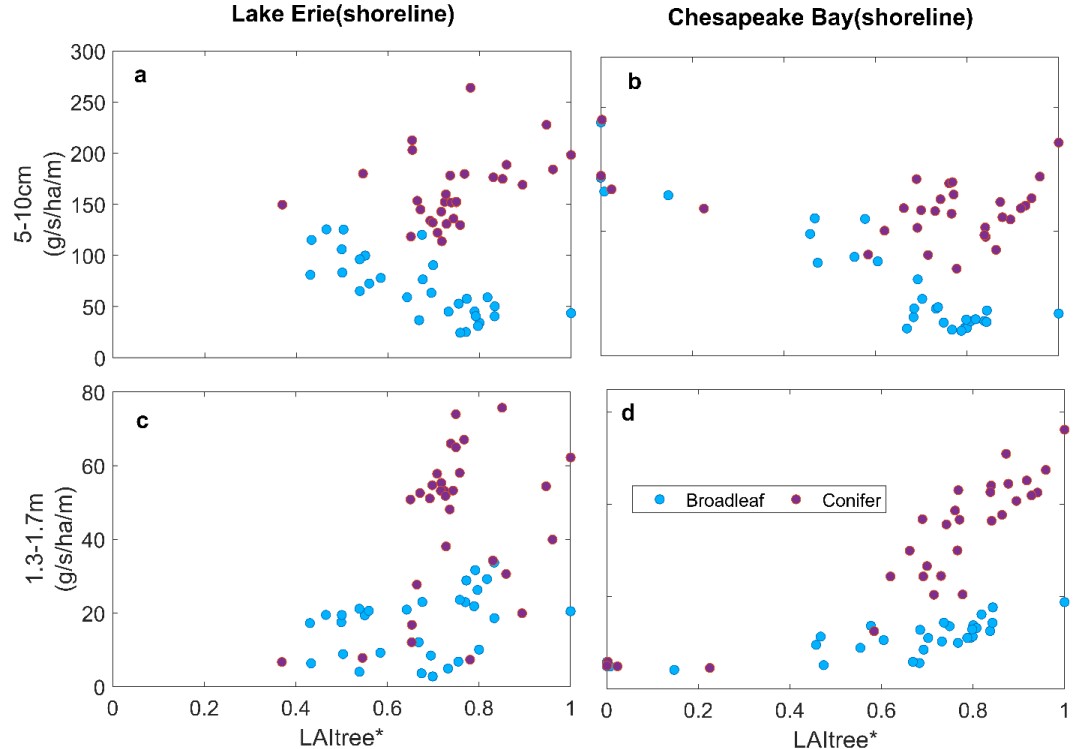