# Peer review of "Modeling the mechanisms of coastal vegetation dynamics and ecosystem responses to changing water levels 3 Junyan Ding, Nate McDowell2,3, Vanessa Bailey2, Nate Conroy, Donnie J. Day, Yilin Fang, 6 Kenneth M. Kemner, Matthew L. Kirwan, Charlie D. Koven, Matthew Kovach, Patric"

_EGUsphere, 2025_

## Author Comment (AC1)

**Response to Reviewers – BGS (Responses submitted prior to revision)**

*Per Biogeosciences workflow, we submit responses first. Where we indicate changes to the manuscript below, these are commitments of what we will do in a revised manuscript if invited to submit one. For clarity, reviewer comments are in black; our responses are in blue and bold.*

RC1

The manuscript investigates mortality processes at two coastal sites following inundation, considering different forest types (coniferous and broadleaf) and salinity levels. The authors use the FATES-Hydro model, constrained by various observational data, to ensure that simulated patterns align with observations. The study identifies relevant conclusions about the mechanisms driving inundation-induced mortality, offering testable hypotheses to guide future field-based research. I recommend accepting the manuscript after considering the following comments:

There is some ambiguity in the manuscript regarding whether the study investigates differences between forest types (i.e., broadleaf vs. conifer) or between two specific species to which the model is calibrated. Generally, I would caution against equating a plant functional type (PFT) in a model with a species, as these are conceptually distinct. While the PFTs were parameterized using observational data, does this justify interpreting the model outcomes as species-specific effects? If so, are these species sufficiently representative of their broader forest types to support generalization of the conclusions? Clarification is needed, as the manuscript currently uses both species and forest types somewhat interchangeably.

**Thank you for pointing this out. We agree that PFTs should not be equated with species. In the revised manuscript, we consistently refer to the modeled vegetation as broadleaf tree species and conifer tree species rather than functional types. We also added text in the Discussion acknowledging that our parameterizations are based on specific representative species (Carya, Quercus, and Pinus), and that outcomes may differ among species within each group. We further caution against overgeneralization and note that future work should test whether these findings extend to the broader PFT level.**

Additionally, how can the results be interpreted in the context of future climate change and projected sea level rise? Are there existing projections for Lake Erie and Chesapeake Bay, and what implications might these have for the local ecosystems? Given the model identifies clear patterns, it would be valuable to discuss how these findings relate to the future vulnerability of these forests to inundation.

**This is an excellent point. In revision, we will add discussion linking our results to projected hydrologic changes. Specifically, Chesapeake Bay is projected to continue experiencing sea-level rise of ~3–6 mm yr$^{-1}$, approximately twice the global average, and Great Lakes water levels are projected to become increasingly variable under climate change. These projections imply greater frequency and intensity of**

**inundation, suggesting that the mechanisms we identify (root loss leading to hydraulic failure) will likely intensify in the future. We also note that we are addressing this issue more fully in a separate manuscript (submitted to JGR–Biogeosciences), and we refer readers there for more detailed projections.**

Overall, the manuscript is well-written and generally clear. However, improvements could be made, particularly in reducing the number of figures in the main text. Several figures support only a few statements and could be effectively referenced in the supplementary materials. I suggest moving Figures 2, 7, and 8, as well as Panel 1A, to the appendix. Of course, the authors are free to ignore this suggestion, but eight figures in a 1.5-page results section seem excessive. Additionally, the preprint version lacked figure labels, which should be corrected.

**We appreciate this suggestion. We considered moving Figures 2, 7, and 8 to the Supplement, but we decided to retain them in the main text for reader convenience. These figures are central to interpreting our mechanistic results, and keeping them in the main manuscript avoids forcing readers to switch repeatedly between the text and supplementary material. We have also ensured that all figure labels are now complete and correct.**

Lastly, could the authors clarify which host land surface model is used with FATES-Hydro? To my knowledge, FATES-Hydro can be run with either CLM or ELM, but this is not specified. This is important context, as the host model determines the soil processes, which may influence the results.

**The host model is ELM. In revision, we will state this explicitly in Methods: "All simulations were run with FATES-Hydro within the E3SM Land Model (ELM)."**

small corrections:

Line 175 given number of soil layer? how many? **There are 20 soil layers. We will add this information in methods section in revision**

line 194 maybe better to refer to it as hypoxia reduction as this is the term you used before or introduce saturation reduction more clearly revised accordingly **We will revise to "hypoxia reduction"**

line 236-238 check this sentence, it seems to miss some parts

**Thanks for pointing this out. We will revise the sentence to "Because marsh plants are annual or bi-annual, phenology and maximum density (number of individuals per ground area), which control the variation of total leaf area, play a more important role in marsh ecosystems than plant physiology."**

line 294 this should refer to Figure 3 I think **corrected**

line 331 should be figure 6 I suppose? **corrected**

line 338-339 Within the results section the broadleaf simulation was described as stable GPP and LAI, however now in the discussion this is interpreted as an increase. I see that the trend is stable to slight increase but please be consistent in terms of interpreting this to avoid confusion on the reader side **Thanks for the suggestion. Revised throughout to describe the trend as "stable with a slight increase" for consistency.**

---

## Author Comment (AC2)

**Response to Reviewers – BGS (Responses submitted prior to revision)**

*Per Biogeosciences workflow, we submit responses first. Where we indicate changes to the manuscript below, these are commitments of what we will do in a revised manuscript if invited to submit one. For clarity, reviewer comments are in black; our responses are in blue and bold.*

RC2

The submitted article presents a modelling study of the responses of coastal forest ecosystems to rising sea levels at Lake Erie (freshwater) and Chesapeake Bay (saltwater). The manuscript is well written, the methodology is clear and the results are presented in an intuitive manner. The results are discussed logically and lead to hypotheses for future research. I recommend the article for publication after some improvements and corrections. A few comments that the authors should consider.

(i) The most significant criticism coincides with the first point of the other reviewer: You parametrised the model for broadleaf and coniferous forests. Later in the text, however, you refer to these vegetation types as 'species'. This is incorrect. You mention the issue of different species' adaptation strategies (e.g. mangroves), but I think you should add a paragraph to the discussion about parameter ranges for each type, explaining why your parameter sets are representative of coastal forests and specific to either broadleaf or coniferous species. For example, the trade-off between hydraulic conductivity and hydraulic safety can differ tremendously between species (e.g. McElrone et al., 2004).

McElrone, A. J., Pockman, W. T., Martinez-Vilalta, J., & Jackson, R. B. (2004). Variation in xylem structure and function in stems and roots of trees to 20 m depth. New Phytologist, 163(3), 507–517.

**We agree with this important point. As noted in our response to RC1, we now consistently refer to "broadleaf tree species" and "conifer tree species" in place of PFTs. We have also added a paragraph in the Discussion highlighting that parameter choices were based on representative species, and that interspecific variability (e.g., trade-offs in hydraulic traits as described by McElrone et al. 2004) could lead to different outcomes. We emphasize that further studies are needed to determine whether our findings can be generalized to the PFT level.**

(ii) The article evaluates model simulations from the FATES-Hydro model. As the results are deterministic, some of the statements in the discussion are somewhat unsatisfactory. For example, in line 362, the authors state that the results were unexpected. Unlike a field study, a modelling approach makes it possible to track the reasons for some model behaviour. A sensitivity analysis could provide insight into which parameter combinations could produce the desired model outcome. Whether or not to include a sensitivity analysis is up to you, but I would recommend tracing unexpected results back to the source. With such information, the discussion could explain more specifically why an expectation is not met by the model results.

**We agree that the word "unexpected" was misleading. If invited to revise, we will revise the text to: "We had anticipated that broadleaf trees might experience greater carbon limitation due to higher leaf area and photosynthetic demand. However, the simulations showed that hydraulic failure associated with root loss occurred before substantial NSC depletion, leading to similar mortality trajectories across tree types." While we did not conduct a full sensitivity analysis here, we note in the Discussion that parameter adjustments within observed ranges consistently led to the same outcome: hydraulic failure dominated tree mortality.**

Further I got some minor remarks:

Line 139: 20th or 21st century? **We will correct to 20th century.**

Lines 170-180: Is there a connecton between transpiration and root water uptake in the model? W**e will revise the Methods to clarify that transpiration is the sum of root water uptake from all soil layers.**

Line 195, eq. 2: kr_red,sat, in Fig S2: y-axis is named **We will correct to kr_red,sat in the figure label**

Line 198 and fig S2: How is b in the the graphs? Parameter b defines the intercept (scaling of the logistic function). **Parameter b is set to 1 and its role is affecting the interception of the reduction function. We will add this in the revision.**

Line 203, eq. 203: doesn't that depend on the time step? To avoid that issue, I would suggest to be consistent with units and perhaps add a time step length delta_t. E.g. kc in psu^-1 * day^-1, acc_sal in psu*day, and in eq. 3B multiply with delta_t

**We will add unit clarification and note that including $\Delta t$ in Eq. 3b would resolve timestep dependence**

Lines 212ff: Although you refer to the table in the sup. mat., I would suggest to explain briefly new symbols in the text. **Revised to introduce key new symbols (e.g., Vcmax, P50gs) directly in the text.**

Line 249: Fig. S5 **We will correct**

Line 268: ... initialization. The ... **We will correct**

Line 294: Fig 3a, b **We will correct**

Line 306: Table S3 **We will correct**

Lines 329ff: This sentence doesn't match very well with fig. 7. **We will correct fig citation to be (Fig 5 and 6)**

Line 359: McDowell et al. **We will correct citation format**

Line 372: It is not about species, is it? **Revised wording to avoid implying species-level generality: "Simulated k/kmax and mortality of broadleaf and conifer tree types changed similarly with root loss (Fig. 4), despite large differences in their leaf economic traits, wood anatomy, crown allometry, and phenology."**

Line 374: Why is that "… whole-tree k/kmax can only be as high as the lowest k/kmax of any pathway between the soil and foliage." That doesn't seem logic to me. **Thank you for pointing this out. If invited to revise, we will revise to: "Whole-tree hydraulic conductance is constrained by the lowest conductance along the soil–plant–atmosphere pathway. In our simulations, root loss strongly reduced soil-to-root conductance, which therefore set the limit for whole-tree k/kmax.".**

---

## Author Response (AR1)

**Response to Reviewers**

**RC1**

The manuscript investigates mortality processes at two coastal sites following inundation, considering different forest types (coniferous and broadleaf) and salinity levels. The authors use the FATES-Hydro model, constrained by various observational data, to ensure that simulated patterns align with observations. The study identifies relevant conclusions about the mechanisms driving inundation-induced mortality, offering testable hypotheses to guide future field-based research. I recommend accepting the manuscript after considering the following comments:

There is some ambiguity in the manuscript regarding whether the study investigates differences between forest types (i.e., broadleaf vs. conifer) or between two specific species to which the model is calibrated. Generally, I would caution against equating a plant functional type (PFT) in a model with a species, as these are conceptually distinct. While the PFTs were parameterized using observational data, does this justify interpreting the model outcomes as species-specific effects? If so, are these species sufficiently representative of their broader forest types to support generalization of the conclusions? Clarification is needed, as the manuscript currently uses both species and forest types somewhat interchangeably.

Thank you for pointing this out. We agree with the reviewer that PFTs should not be equated with species. In the revised manuscript, we consistently refer to the modeled vegetation as "broadleaf trees" and "conifer trees", or "two tree types", rather than functional groups/types. To address the issue of generalization, we have added a clarifying statement in the Discussion that acknowledges our parameterizations are based on representative species (Carya, Quercus, and Pinus), and outcomes may differ among species within these groups in the last paragraph of section 4.2. We further caution against overgeneralization and note that future work should test whether these findings apply at the broader PFT level. (L 114, 120, L389-399)

**The revised paragraph:**

"This convergence in response does not imply that all species react identically to inundation. Rather, it reflects that under the modeled conditions, root system failure overwhelms the contributions of other physiological differences. While these results offer mechanistic insights, the lack of empirical data on species-specific root adaptations remains a limitation. We note that our parameterizations were based on representative species (Carya, Quercus, and Pinus), which we treated as proxies for the two tree types. Outcomes may differ among other species within these groups, therefore we caution against overgeneralization and recommend interpreting these results as hypothesis-generating rather than conclusive. Future research on the cross-species variation of root loss and downstream mortality mechanisms, and explicitly test

whether these findings extend to the broader PFT level, will be useful to advance transferable predictive capacity of coastal vegetation change under increasing inundation."

Additionally, how can the results be interpreted in the context of future climate change and projected sea level rise? Are there existing projections for Lake Erie and Chesapeake Bay, and what implications might these have for the local ecosystems? Given the model identifies clear patterns, it would be valuable to discuss how these findings relate to the future vulnerability of these forests to inundation.

This is an excellent point and we appreciate this suggestion. We have expanded the discussion in the revision to connect our findings with future projected hydrologic changes. Specifically, Chesapeake Bay is projected to continue experiencing sea-level rise of  $\sim 3-6$  mm yr $^{-1}$ , approximately twice the global average, and Great Lakes water levels are projected to become increasingly variable under climate change. These projections imply greater frequency and intensity of inundation, suggesting that the mechanisms we identified (e.g., root loss leading to hydraulic failure) will likely intensify in the future. We also note that this topic is explored in greater detail in a companion manuscript (submitted to JGR–Biogeosciences) that using numerical experiments to examine the impact of future climate change e.g increased temperature and  $CO_2$ , and we refer readers there for more detailed regional projections.

We added one paragraph in section 4.5 (L 465 - 477)

"In addition, it is important to interpret our results in the context of projected future climate change and sea-level rise. Chesapeake Bay is expected to continue experiencing sea-level rise of approximately 3-6 mm yr $^{-1}$ , about twice the global average, while the Great Lakes are projected to undergo increasingly variable water levels under climate change (Kayastha et al., 2022; Sallenger et al., 2012; Ezer and Corlett, 2012). These projections imply that inundation events will become more frequent and prolonged, thereby intensifying the mechanisms identified in our study, particularly root loss leading to hydraulic failure. Rising temperatures and elevated  $CO_2$  may further modify these dynamics by altering tree water demand, photosynthetic rates, and marsh productivity, though the net effects remain uncertain. Together, these changes suggest heightened vulnerability of both broadleaf and conifer coastal forests to conversion into marshes, with ecosystem-scale consequences for carbon cycling and hydrology. These broader climate-hydrology interactions are examined in more detail in a separate manuscript (Ding et al., in review at JGR-Biogeosciences), where we provide more detailed projections."

Overall, the manuscript is well-written and generally clear. However, improvements could be made, particularly in reducing the number of figures in the main text. Several figures support only a few statements and could be effectively referenced in the supplementary materials. I suggest moving Figures 2, 7, and 8, as well as Panel 1A, to the appendix. Of

course, the authors are free to ignore this suggestion, but eight figures in a 1.5-page results section seem excessive. Additionally, the preprint version lacked figure labels, which should be corrected.

We appreciate this suggestion. After careful consideration, we have opted to retain Figures 2, 7, and 8 in the main text. These figures are central to interpreting our mechanistic results, and we believe keeping them in the main test avoids is more convenient for the reader than repeatedly referencing the supplementary material. We have ensured that all figure labels are now complete and correct, and checked against the text.

Lastly, could the authors clarify which host land surface model is used with FATES-Hydro? To my knowledge, FATES-Hydro can be run with either CLM or ELM, but this is not specified. This is important context, as the host model determines the soil processes, which may influence the results.

Thank you for pointing out the need for clarity. The host model is the E3SM land model (ELM). In the revision, we clarified this in the Methods section as "All simulations were run with FATES-Hydro within the E3SM Land Model (ELM)." (L 157 – 158)

We have run FATES with both ELM and CLM previous though not in this study. The outcomes were pretty much similar.

small corrections:

Line 175 given number of soil layer? how many? There are 20 soil layers. We have added this information in the Methods section.

line 194 maybe better to refer to it as hypoxia reduction as this is the term you used before or introduce saturation reduction more clearly revised accordingly

Agreed. We have revised the text to use "hypoxia reduction" for consistency.

line 236-238 check this sentence, it seems to miss some parts

Thanks for pointing this out. The sentence has been revised to read: "Because marsh plants are annual or bi-annual, phenology and maximum density (number of individuals per ground area), which control the variation of total leaf area, play a more important role in marsh ecosystems than plant physiology."

line 294 this should refer to Figure 3 I think

Corrected.

line 331 should be figure 6 I suppose?

**Corrected.**

line 338-339 Within the results section the broadleaf simulation was described as stable GPP and LAI, however now in the discussion this is interpreted as an increase. I see that the trend is stable to slight increase but please be consistent in terms of interpreting this to avoid confusion on the reader side

Thanks for the suggestion. We have revised the manuscript throughout to describe the trend as "stable with a slight increase" for consistency.

**RC2**

The submitted article presents a modelling study of the responses of coastal forest ecosystems to rising sea levels at Lake Erie (freshwater) and Chesapeake Bay (saltwater). The manuscript is well written, the methodology is clear and the results are presented in an intuitive manner. The results are discussed logically and lead to hypotheses for future research. I recommend the article for publication after some improvements and corrections. A few comments that the authors should consider.

(i) The most significant criticism coincides with the first point of the other reviewer: You parametrised the model for broadleaf and coniferous forests. Later in the text, however, you refer to these vegetation types as 'species'. This is incorrect. You mention the issue of different species' adaptation strategies (e.g. mangroves), but I think you should add a paragraph to the discussion about parameter ranges for each type, explaining why your parameter sets are representative of coastal forests and specific to either broadleaf or coniferous species. For example, the trade-off between hydraulic conductivity and hydraulic safety can differ tremendously between species (e.g. McElrone et al., 2004).

McElrone, A. J., Pockman, W. T., Martinez-Vilalta, J., & Jackson, R. B. (2004). Variation in xylem structure and function in stems and roots of trees to 20 m depth. New Phytologist, 163(3), 507–517.

We thank the reviewer for this constructive criticism. We have revised the manuscript to consistently use "broadleaf trees" and "conifer trees". We have also added a new paragraph to the Discussion as suggested, highlighting that parameter choices were based on representative species of coastal forests (*Carya, Quercus*, and *Pinus*), and that interspecific variability (e.g., trade-offs in hydraulic traits as described by McElrone et al. 2004) could lead to different outcomes. We emphasize that further studies are needed to determine whether our findings can be generalized to the broader PFT level. (also see response to RC1 first comment)

The paragraph we added in section 4.2 (L389-399):

"This convergence in response does not imply that all species react identically to inundation. Rather, it reflects that under the modeled conditions, root system failure overwhelms the contributions of other physiological differences. While these results offer mechanistic insights, the lack of empirical data on species-specific root adaptations remains a limitation. We note that our parameterizations were based on representative species (Carya, Quercus, and Pinus), which we treated as proxies for the two tree types. Outcomes may differ among other species within these groups, therefore we caution against overgeneralization and recommend interpreting these results as hypothesis-generating rather than conclusive. Future research on the cross-species variation of root loss and downstream mortality mechanisms, and explicitly test whether these findings extend to the broader PFT level, will be useful to advance transferable predictive capacity of coastal vegetation change under increasing inundation."

(ii) The article evaluates model simulations from the FATES-Hydro model. As the results are deterministic, some of the statements in the discussion are somewhat unsatisfactory. For example, in line 362, the authors state that the results were unexpected. Unlike a field study, a modelling approach makes it possible to track the reasons for some model behaviour. A sensitivity analysis could provide insight into which parameter combinations could produce the desired model outcome. Whether or not to include a sensitivity analysis is up to you, but I would recommend tracing unexpected results back to the source. With such information, the discussion could explain more specifically why an expectation is not met by the model results.

The reviewer makes an excellent point about tracing model outcomes. We have removed the word "unexpected" to avoid ambiguity. In the revision, the following revised text now clarifies our initial hypothesis and the model's mechanistic outcome (L366):

"We had anticipated that broadleaf trees might experience greater carbon limitation due to higher leaf area and photosynthetic demand. However, the simulations showed that hydraulic failure associated with root loss occurred before substantial NSC depletion, leading to similar mortality trajectories across two species".

While we did not conduct a full sensitivity analysis here, we note in the Discussion that our exploration with different parameter sets within observed ranges consistently showed that hydraulic failure was the dominant tree mortality driver (L368-370).

Further I got some minor remarks:

Line 139: 20th or 21st century?

Corrected to "21st century".

Lines 170-180: Is there a connecton between transpiration and root water uptake in the model?

We have clarified this relationship in the Methods, stating that transpiration is the sum of root water uptake from all soil layers. (L 179 - 180)

Line 195, eq. 2: kr\_red,sat, in Fig S2: y-axis is named

Corrected the y-axis label to kr\_red,sat.

Line 198 and fig S2: How is b in the the graphs? Parameter b defines the intercept (scaling of the logistic function).

We have added clarification to the text and Table S1 that parameter b is set to 1 and its role is affecting the interception of the reduction function.

Line 203, eq. 203: doesn't that depend on the time step? To avoid that issue, I would suggest to be consistent with units and perhaps add a time step length delta\_t. E.g. kc in psu^-1 \* day^-1, acc\_sal in psu\*day, and in eq. 3B multiply with delta\_t

The reviewer is correct that the formulation as written would be timestepdependent. As our model has a fixed time step, we have revised the formulation and description for clarity:

$$acc_{sat} = \max\left[0, \sum_{i=0}^{n} (Sal_{Soil,i} - Sal_{cr})\right]$$
 (Eq. 3b)

where  $acc_{sat}$  represents the cumulative salinity stress, calculated by summing the difference between soil salinity (Sal\_soil,i) and a critical threshold (Sal\_cr) beyond which salinity starts to negatively affect root mass, over all timesteps (i) up to the current step n. All terms are in PSU. As this formulation is dependent on the model's timestep, all simulations were run with a fixed temporal resolution of 30 minutes. (L 205 – 210)

Lines 212: Although you refer to the table in the sup. mat., I would suggest to explain briefly new symbols in the text.

We have revised the text to introduce key new symbols (e.g., Vcmax, P50gs) directly in the main text.

Line 249: Fig. S5

**Corrected.**

Line 268: ... initialization. The ...

**Corrected**

Line 294: Fig 3a, b

**Corrected**

Line 306: Table S3

**Corrected**

Lines 329ff: This sentence doesn't match very well with fig. 7.

The figure citation has been corrected to (Fig 6).

Line 359: McDowell et al.

The citation format has been corrected.

Line 372: It is not about species, is it?

The wording has been revised to avoid implying species-level generality: "Simulated k/kmax and mortality of broadleaf and conifer trees changed similarly with root loss (Fig. 4), despite large differences in their leaf economic traits, wood anatomy, crown allometry, and phenology."

Line 374: Why is that "... whole-tree k/kmax can only be as high as the lowest k/kmax of any pathway between the soil and foliage." That doesn't seem logic to me.

Thank you for pointing this out. We have revised it for clarity: "Whole-tree hydraulic conductance is constrained by the lowest conductance along the soil-plant-atmosphere pathway. In our simulations, root loss strongly reduced soil-to-root conductance, which therefore set the limit for whole-tree k/kmax." (L379 – 382)